# Addition of Olive Leaf Extract to a Mixture of Algae and Extra Virgin Olive Oils Decreases Fatty Acid Oxidation and Synergically Attenuates Age-Induced Hypertension, Sarcopenia and Insulin Resistance in Rats

**DOI:** 10.3390/antiox10071066

**Published:** 2021-07-01

**Authors:** Daniel González-Hedström, María de la Fuente-Fernández, Teresa Priego, Ana Isabel Martín, Sara Amor, Asunción López-Calderón, Antonio Manuel Inarejos-García, Ángel Luís García-Villalón, Miriam Granado

**Affiliations:** 1Departamento de Fisiología, Facultad de Medicina, Universidad Autónoma de Madrid, 28029 Madrid, Spain; dgonzalez@pharmactive.eu (D.G.-H.); maria.delafuente@uam.es (M.d.l.F.-F.); sara.amor@uam.es (S.A.); angeluis.villalon@uam.es (Á.L.G.-V.); 2Pharmactive Biotech Products S.L. Parque Científico de Madrid, Avenida del Doctor Severo Ochoa, 37 Local 4J, 28049 Madrid, Spain; aminarejos@hotmail.com; 3Departamento de Fisiología, Facultad de Medicina, Universidad Complutense de Madrid, 28029 Madrid, Spain; tpriegoc@med.ucm.es (T.P.); anabelmartin@med.ucm.es (A.I.M.); alc@med.ucm.es (A.L.-C.); 4CIBER Fisiopatología de la Obesidad y Nutrición, Instituto de Salud Carlos III, 28029 Madrid, Spain

**Keywords:** aging, olive leaf extract, omega 3 fatty acids, algae oil, extra virgin olive oil, insulin resistance, cardiovascular, inflammation, oxidative stress, endothelial dysfunction, hypertension

## Abstract

Olive-derived products, such as virgin olive oil (EVOO) and/or olive leaf extracts (OLE), exert anti-inflammatory, insulin-sensitizing and antihypertensive properties and may be useful for stabilizing omega 3 fatty acids (n-3 PUFA) due to their high content in antioxidant compounds. In this study, the addition of OLE 4:0.15 (*w**/w*) to a mixture of algae oil (AO) rich in n-3 PUFA and EVOO (25:75, *w*/*w*) prevents peroxides formation after 12 months of storage at 30 °C. Furthermore, the treatment with the oil mixture (2.5 mL/Kg) and OLE (100 mg/Kg) to 24 month old Wistar rats for 21 days improved the lipid profile, increased the HOMA-IR and decreased the serum levels of miRNAs 21 and 146a. Treatment with this new nutraceutical also prevented age-induced insulin resistance in the liver, gastrocnemius and visceral adipose tissue by decreasing the mRNA levels of inflammatory and oxidative stress markers. Oil mixture + OLE also attenuated the age-induced alterations in vascular function and prevented muscle loss by decreasing the expression of sarcopenia-related markers. In conclusion, treatment with a new nutraceutical based on a mixture of EVOO, AO and OLE is a useful strategy for improving the stability of n-3 PUFA in the final product and to attenuate the cardiometabolic and muscular disorders associated with aging.

## 1. Introduction

In the next decades the world population is expected to double or even triplicate, so the great incidence of aging-associated co-morbidities will represent a major burden for the healthcare systems [1]. Thus, the search of new strategies to prevent these diseases, with ideally less side effects than the conventional pharmacological treatments, is highly desirable. In this regard, dietary interventions are becoming popular candidates due to their higher accessibility and fewer side effects [2,3]. Among them, the use of omega-3 polyunsaturated fatty acids (n-3 PUFA) from marine products (mainly fish) and the Mediterranean diet, which includes a high consumption of products derived from the olive tree (*Olea europea L.*), mainly olive oil [4], stand out for their beneficial effects. Both interventions have shown health benefits in the elderly population at the metabolic and cardiovascular level, demonstrating great anti-inflammatory and antioxidant properties [4,5,6].

The Mediterranean diet promotes longevity [5] due, at least in part, to its content in antioxidant compounds [7] and its beneficial effects on cardiovascular function [8]. Furthermore, the regular consumption of extra virgin olive oil (EVOO) is associated with lower risk of suffering metabolic syndrome [9] and improves its symptoms in overweight and obese aged adults [10] and in menopausal women [11]. The beneficial effects of EVOO are due to the presence of different compounds, such as oleic acid which is a monounsaturated fatty acid that exerts several beneficial effects on cardiovascular function, improving diastolic function [12], reducing cardiovascular insulin resistance and improving the atherogenic process [13]. However, some of the positive effects of EVOO on metabolism and cardiovascular function are also attributed to polyphenols, such as tyrosol and hydroxytyrosol [14]. These compounds are also present in olive leaves and are reported to exert several beneficial health effects such as anti-inflammatory, anti-obesity, cardioprotective, hypocholesterolemic, anti-hypertensive and hypoglycemic [15,16] due to their potent antioxidant actions [17]. Moreover, the great antioxidant capacity of phenolic compounds from olive leaf extracts (OLE) renders them as attracting ingredients from a technological point of view, since they can be used to reduce oxidation processes in different formulations such as functional foods and nutraceutical products [18,19].

Similar to EVOO and OLE, the consumption of n-3 PUFA from fish also exerts beneficial cardiometabolic effects throughout life [6] by improving insulin resistance [20] and vascular function in healthy [21], hypercholesterolemic [22], old [23] and type-2 diabetic subjects [24]. However, due to the high demand, the consumption of fish products rich in n-3 PUFA is not sustainable in the long term [25], which then requires the necessary use of alternative and more ecological sources such as algae oils (AO) that are also very rich in these fatty acids [26]. Independent of their origin, supplements rich in n-3 PUFA require addition of antioxidants [27] due to their susceptibility to oxidation which results in the formation of oxidation products with potential toxic effects that may limit their health benefits [28]. Thanks to its high content in phenolic compounds, it has been suggested that both EVOO and OLE can be used as antioxidants; they may be good candidates for stabilizing n-3 PUFA [29].

Arterial hypertension is the most frequent chronic disease in the elderly, with more than 70% of both men and women over the age of 65 suffering from this condition [30]. Age-induced hypertension is the result of the stiffness of the conducting arteries, mainly the aorta [31], due to structural alterations of the vascular walls that include increase in collagen, loss of elastin and calcification and derangement of the fibres [32]. Another factor that contributes to the development of age-induced hypertension is endothelial dysfunction, which is a condition associated with an imbalance in the production of vasodilating agents, such as nitric oxide (NO), and vasoconstrictor agents such as endothelin-1 (ET-1) by the vascular endothelium that results in impaired vasodilation [33].

Aging is also associated with profound metabolic changes which include a reduction in tolerance to carbohydrates [34]. In addition to the progressive loss of the pancreatic capacity to produce insulin [35], numerous studies have shown that carbohydrate intolerance associated with aging is mainly due to insulin resistance [36]. Insulin resistance contributes to the development of cardiovascular alterations through impaired lipid uptake and storage, promoting increased plasma levels of low density lipoprotein (LDL), which is typical of this syndrome [36]. Multifactorial mechanisms underly the development of insulin resistance, with visceral adiposity, abnormal free fatty acid metabolism and several hormones and regulatory factors being involved in this process [2]. It is characterized by an impaired activation of the phosphoinositide 3–kinase (PI3K)/Akt pathway, which results in reduced glycogen synthesis/storage as well as impaired glucose uptake by fat and skeletal muscle cells promoting hyperglycaemia [36]. In skeletal muscles, the age-induced decrease in insulin sensitivity is also associated with the development of sarcopenia, a condition that is characterized by a gradual loss of muscle mass which results in decreased muscle strength resulting in fragility and reduced mobility [37,38].

Age-induced insulin resistance not only affects metabolic tissues but also the cardiovascular system [39,40]. In vascular endothelial cells, nitric oxide production is reduced due to a decrease in the PI3K/Akt pathway activation. However, insulin-induced activation of the mitogen activated kinases (MAPK) pathway is not affected by insulin resistance, which results in the thickening of vascular wall and increased production of the vasoconstrictor endothelin-1 by the endothelium contributing to the development of hypertension [41,42].

Inflammation and oxidative stress are common mechanisms underlying the development of both metabolic and vascular insulin resistance as well as endothelial dysfunction associated with aging [33]. As a result of increased visceral adiposity, aging is associated with a state of low-grade chronic inflammation produced by increased serum levels of pro-inflammatory cytokines. Pro-inflammatory cytokines act in a paracrine and autocrine manner in peripheral tissues inducing insulin resistance through impaired tyrosine phosphorylation of insulin receptor substrate (IRS)-1 [43]. On the other hand, the development of insulin resistance is also highly associated with oxidative stress, a phenomenon characterized by the increased production of reactive oxidative species (ROS) [44]. In aging, the high levels of ROS promote that glucose transporter type 4 (GLUT-4) storage vesicles are transported to lysosomes for degradation instead of the plasma membrane dealing with hyperglycaemia [44]. ROS are the result of both increased activity of prooxidant enzymes such as nicotinamide adenine dinucleotide phosphate (NADPH) oxidase 4, which is an enzyme in humans that is encoded by the gene NOX-4, and the result of the impaired activity of enzymes that act as radical scavengers, such as catalase, glutathione peroxidases (GPx) and superoxide dismutase (SOD-1) [45].

A previous study from our group has demonstrated that addition of EVOO to AO in a proportion 3:1 not only attenuates the cardiometabolic alterations associated with aging [46] but also improves n-3 PUFA oxidative stability enlarging their shelf-life [47]. However, the possible synergistic biological effects of AO + EVOO and OLE in age-induced cardiometabolic alterations, as well as the ability of OLE to decrease n-3 PUFA oxidation has not been studied yet. Thus, the first objective of this study is to evaluate if the addition of OLE to a mixture of EVOO and AO has protective effects against omega 3 fatty acid oxidation induced by storage. In addition, since it is reported that the simultaneous use of different natural compounds may have synergistic effects [48,49], the second objective of this work is to analyze if the addition of OLE to a mixture between AO and EVOO could improve their beneficial effects to alleviate the cardiovascular and metabolic alterations related to aging.

## 2. Materials and Methods

### 2.1. Materials

Pharmactive Biotech Products S.L. (Madrid, Spain) provided the powdered water-soluble samples of OLE (*Olea europaea* L.) standardized to 1 mg/g of luteolin-7-O-glucoside by HPLC and to 30% of ortho-diphenols by UV/Vis. They were all stored in darkness until their addition into the feeding bottles. EVOO of Cornicabra variety with 80% of oleic acid and 60 mg/g of secoiridoids was obtained from Aceites Toledo S.A. (Los Yébenes, Spain) and AO (*Schizochytrium* spp: 35% docosahexaenoic acid (DHA), 20% eicosapentaenoic acid (EPA) and 5% docosapentaenoic acid (DPA) was obtained from DSM (Heerlen, Netherlands).

Water was purified using a Milli-Q Millipore system (Bedford, MA, USA) and methanol and acetonitrile were of HPLC grade and purchased from Scharlab (Barcelona, Spain). All other substances and solvents were obtained from LabBox (Madrid, Spain).

### 2.2. Study of the Oxidative Stbaility of the Oil Mixture in the Presence or Absence of the OLE

#### 2.2.1. Oil Mixture Preparation

In order to obtain the oil mixture of AO and EVOO, AO:EVOO were mixed in an axial flow stirrer for 15 min in a proportion 25:75 (*w**/w*).

The sample of the oil mixture and the OLE was prepared in a proportion of 4:0.15 (*w**/w*) using an axial flow stirrer for 15 min.

#### 2.2.2. Oxidative Stability Conditions

Oil mixtures of 10 g samples in the presence/absence of OLE were stored in unsealed transparent glass vials with the bottleneck space of 1 mL free of sample and kept at 30 °C in darkness for 0, 1, 3, 6, 9 and 12 months. Three vials were frozen (−20 °C) and kept in darkness at the end of each time point until further analyses.

#### 2.2.3. Fatty Acid Composition

According to the European Regulation (EEC 2568/91), the composition of fatty acids of the mixtures was determined as previously described by gas chromatography (GC) [50]. Briefly, the chromatographic separation was performed using a Shimadzu GC-2025 gas chromatograph (Shimadzu; Kioto, Japan). This chromatograph was equipped with a ZB-FFAP capillary column (50 m × 0.32 mm × 0.50 µm) (Phenomenex; Torrance, CA, USA), an AOC-20i autosampler (Shimadzu; Kioto, Japan) and a FID detector. Supelco 37 Component FAME Mix (Merk; Dramstadt, Germany) was used as standardized fatty acid mix to identify the peaks of each fatty acid within the samples. Percentage (%) of each fatty acid peak with respect to total quantified oil was used to express the results.

#### 2.2.4. Oxidation Indexes

Oil oxidation parameters as Peroxide Value (PV) for primary oxidation products and p-Anisidine Value (AV) for secondary oxidation products were measured as previously described [47]. PV was measured as indicated in the European Regulations (EEC 2568/91) and expressed as milliequivalents of active oxygen per kilogram of oil sample (mEquiv O_2_/Kg oil). AV was measured as indicated in the analytical methods of the British Standard method (BS 684 1998).

#### 2.2.5. Analysis of Phenolic Compounds by HPLC

##### Extraction of Phenolic Compounds

A 10 g sample of the mixture was mixed with 1 mL of methanol/water (60:40 *v*/*v*) with approximately 225 mg/mL of syringic acid (Sigma-Aldrich, St. Louis, MO, USA) to used it as an internal standard [51]. Later, 5 mL of n-hexane and 2 mL of methanol/water (60:40 *v*/*v*) were added and mixed, and a centrifugation at 4000 rpm for 5 min was performed to remove n-hexane phase. The extract was washed twice with n-hexane, obtaining a final volume of 3 mL of an extract of oil phenolic compounds.

##### HPLC Determination of Phenolic Fraction

The phenolic fraction was determined by high performance liquid chromatography as previously described [47] (HPLC) using a modified method of Mateos R. et al. [52]. Each secoiridoid peak was quantified at 280 nm using syringic acid as the internal standard.

### 2.3. In Vivo Study

#### 2.3.1. Animals

Three months old (Young; *n* = 11) and twenty-four months old (Old; *n* = 17) male Wistar rats were fed ad libitum with standard chow and housed under controlled conditions of temperature (22–24 °C) and humidity (50–60%). All the experiments and handling of animals were approved by the Animal Care and Use Committee of the Universidad Autónoma de Madrid and the Autonomic Government (PROEX 048/18) and conducted according to the European Union Legislation.

#### 2.3.2. Treatment

Half of the old rats (Old) and young rats were administered with water for 21 days while the other half of old rats were (Old + Oil Mixture + OLE) administered with OLE in drinking water and treated orally with 2.5 mL/Kg of the Oil Mixture (75% of EVOO and 25% of AO) once daily. In order to avoid degradation, OLE was renewed every 3 days and the dosage was adjusted to the liquid intake and the animal’s body weights.

Over the three week treatment, body weight was measured daily and food intake and water intake were measured weekly. After the treatment period and an overnight fasting, the animals were killed by decapitation after an overdose of sodium pentobarbital (100 mg/Kg). Trunk blood was collected and the serum was obtained after centrifugation at 3000 rpm for 20 min. Glycemia was measured before sacrifice by venous tail puncture using the glucometer Glucocard G™ (Arkray Factory, Inc.; Shiga, Japan). Immediately after, the subcutaneous (lumbar), visceral (epididymal), perivascular (aortic) and brown (interscapular) adipose tissue depots, as well as the heart, liver, spleen, kidneys, adrenal glands and gastrocnemius and soleus muscles were dissected, weighed and stored at −80 °C for further analysis.

#### 2.3.3. Measurement of Mean Arterial Pressure in Conscious Rats by Tail-Cuff System

One week before sacrifice, mean arterial blood pressure (MAP) was measured by tail-cuff plethysmography as previously described [53]. Values are the average of 5–6 measurements per animal.

#### 2.3.4. Serum Measurements

Serum concentrations of leptin, insulin and adiponectin were determined by ELISA kits according to manufacturer’s instructions (Merck Millipore, Dramstadt, Germany). Likewise, the serum levels of tumor necrosis factor alpha (TNF-α) and Interleukin-6 (IL-6) were analyzed by ELISA (Cusabio, Wuhan, China). Finally, the circulating levels of total cholesterol, low-density lipoprotein (LDL), high-density lipoprotein (HDL), triglycerides and total lipids were determined using colorimetric assays from Spin React (Sant Esteve de Bas, Girona, Spain).

#### 2.3.5. Serum Lipid Extraction and Fatty Acid Analysis

In order to isolate Fatty Acids (FA) from the serum, a lipid extraction process was performed as described by Drews et al. [54]. The FA profile was determined by GC as previously described [46]. Saturated fatty acids (SFA) levels were obtained by the sum of palmitic, palmitoleic and stearic acids. Likewise, n-3 PUFA levels were calculated by the sum of α-linolenic acid (ALA) and EPA, DHA and monounsaturated fatty acids (MUFA) levels were calculated by the sum of oleic and linoleic acids.

#### 2.3.6. Experiments of Vascular Reactivity

The aorta was placed in sterile cold saline solution (NaCl 9 g/L) and cut into 2 mm segments. Afterwards, two steel wires (100 µm) were passed through the lumen of the segments. One of the wires was fixed to a 4 mL organ bath chamber filled with modified Krebs-Henseleit solution (NaCl 115 mM; KCl 4.6 mM; KH_2_PO_4_ 1.2 mM; MgSO_4_ 1.2 mM; CaCl_2_ 2.5 mM; NaHCO_3_ 25 mM; glucose 11 mM) at 37 °C and pH of 7.3–7.4 The other one was connected to a strain gauge for isometric tension recording using a PowerLab data acquisition system (AD Instruments; Colorado Springs, CO, USA). After the setting process, an optimal passive tension of 1 g was applied to each segment. Then, segments were allowed to equilibrate for 60–90 min minutes. Finally, segments were stimulated with potassium chloride (KCl; 100 mM) to measure their ability to contract. Segments were discarded if they failed to contract at least 0.5 g to KCl.

Abdominal aortic segments were used to record the vasoconstrictor response to accumulative doses of noradrenaline (10^−9^–10^−4^ M). Results were expressed as percentage of the contraction to KCl 100 mM.

Thoracic aortic segments precontracted with phenylephrine 10^−7.5^ M were used to study the vasodilator response to cumulative doses of sodium nitroprusside (10^−9^–10^−5^ M), acetylcholine (10^−9^–10^−4^ M) or insulin (10^−8^–10^−5^ M). Relaxation was expressed as a percentage of the final tone after sodium nitroprusside (10^−5^ M) stimulation.

All drugs were obtained from Sigma-Aldrich (St. Louis, MO, USA).

#### 2.3.7. Incubation of Gastrocnemius Muscle and Epididymal Adipose Tissue Explants and Aorta Segments in Presence/Absence of Insulin (10^−7^ M)

Explants of gastrocnemius muscle and epididymal adipose tissue (100 mg) as well as segments of the thoracic aorta were incubated at 37 °C in an incubator (95% O_2_ + 5% CO_2_) with 1.5 mL of Dulbecco’s Modified Eagle’s Medium and Ham’s F-12 medium (DMEM/F-12) supplemented with penicillin (100 U/mL), streptomycin (100 μg/mL) (Invitrogen, Carlsbad, CA, USA) and with glutamine (Gibco, 1:1 mix; Invitrogen, Carlsbad, CA, USA), in the presence/absence of insulin (10^−7^ M) (Sigma-Aldrich, St. Louis, MO, USA). Aorta segments were incubated for 30 min and adipose tissue and gastrocnemius explants for 15 min. Afterwards, both the media and the tissues were collected and stored at −80 °C for further analysis.

#### 2.3.8. Nitrite and Nitrate Concentrations in the Culture Medium

The culture medium from the aorta segment incubations was used to quantify the nitrite and nitrate concentrations by a modified method of the Griess assay [55]. Briefly, one hundred microliters of culture medium were mixed with 100 μL of vanadium chloride. Immediately after one hundred microliters of Griess reagent was added, absorbance was measured at 540 nm after incubation at 37 °C for 30 min. Nitrite and nitrate concentrations were calculated using a NaNO_2_ standard curve and expressed as micromoles per liter.

#### 2.3.9. Protein Quantification by Western Blot

The amounts of 100 mg of gastrocnemius muscle, liver and epididymal adipose tissue were homogenized using radioimmunoprecipitation assay (RIPA) buffer. After centrifugation (12.500 rpm; 4 °C, 20 min), the content of total proteins in the supernatants was quantified by the Bradford method (Sigma-Aldrich; St. Louis, MO, USA) [56]. Electrophoresis was performed using resolving gels with SDS acrylamide (10%) (Bio-Rad; Hercules, CA, USA) (100 µg sample/well) and proteins were then transferred to polyvinylidine difluoride (PVDF) membranes (Bio-Rad; Hercules, CA, USA). After assessing transfer efficiency by Ponceau red dyeing (Sigma-Aldrich; St. Louis, MO, USA), membranes were blocked for 1 h either with 5% (*w/v*) bovine serum albumin (BSA) dissolved inTris-buffered saline (TBS) (for phosphorylated proteins) or with TBS + 5% (*w/v*) non-fat dried milk (for non-phosphorylated proteins). Membranes were then incubated overnight with the primary antibodies: Akt (1:1000; Merck Millipore; Dramstadt, Germany), p-Akt (Ser473) (1:500; Cell Signaling Technology; Danvers, MA, USA), GSK3β (1:1000; Cell Signaling Technology; Danvers, MA, USA) and p-GSK3β (1:1000; Cell Signaling Technology; Danvers, MA, USA). After four 10 min washes, a 90 min incubation with the secondary antibody (1:2000; Pierce; Rockford, IL, USA) conjugated with peroxidase was performed. Finally, membranes were washed several times and chemiluminescence was measured using the BioRad Molecular Imager ChemiDoc XRS System (Hércules, CA, USA). Data are referred to % of control young rats value on each gel.

#### 2.3.10. RNA Extraction and Purification

Total RNA was extracted from epididymal adipose tissue, gastrocnemius, liver, heart and arterial tissues by the Tri-Reagent protocol [57]. After RNA quantification using a Nanodrop 2000 (Thermo Fisher Scientific, Hampton, NH, USA), 1 µg of total RNA was retrotranscribed into cDNA using high capacity cDNA reverse transcription kit (Applied Biosystems; Foster City, CA, USA).

#### 2.3.11. Quantitative Real-Time PCR

Taqman Assays (Applied Biosystems, Foster City, CA, USA) were used to determine the gene expression of cyclooxigenase-2 (COX-2), inducible Nitric Oxide Synthase (iNOS), interleukins 1β (IL-1β), -6 (IL-6) and- 10 (IL-10), tumoral necrosis factor alpha (TNF-α), glutathione reductase (GSR) and peroxidase (GPx); NADPH oxidases 1 (NOX-1) and 4 (NOX-4), superoxide dismutase-1 (SOD1) and lipoxygenase (Alox5) were assessed in the liver, gastrocnemius, white adipose tissue (WAT) and arterial and myocardial tissues by quantitative real-time polymerase chain reaction (qPCR). mRNA levels of glucokinase (GCK) and glycogen synthase kinase 3β (GSK3β) were also determined in the liver and mRNA levels of histone deacetylase 4 (HDAC-4), myogenin, myoblast determination protein 1 (MyoD), atrogin, muscle RING-finger protein-1 (MuRF1), myosin heavy chain I (MHC-I) myosin heavy chain IIa (MHC-IIa), insulin growth factor 1 (IGF-1), peroxisome proliferator activated receptor α (PPARα), PPARγ coactivator 1α (PGC-1α) and BCL2/adenovirus E1B 19 kDa protein-interacting protein 3 (Bnip-3) were assessed in the gastrocnemius. In the WAT, the mRNA levels of lipoprotein lipase (LPL), hormone-sensitive lipase (HSL), fatty acid synthase (FASN) and peroxisome proliferator activated receptor γ (PPARγ) were also measured. Amplification was performed by using the TaqMan Universal PCR Master Mix (Applied Biosystems, Foster City, CA, USA) in a Step One machine (Applied Biosystems, Foster City, CA, USA). In order to normalize the values, 18S was used as a housekeeping gene. Relative expression levels were determined by the ∆∆C_T_ method [58] according to the manufacturer’s guidelines.

#### 2.3.12. Isolation and qRT-PCR of Micro-RNAs from Serum

Micro-RNAs from serum samples were isolated using Qiagen miRNeasy Serum/Plasma Kit (Qiagen; Hilden, Germany) following the instructions of the manufacturer. The quality of RNA was determined using NanoDrop2000 (Thermo Fisher Scientific; Hampton, NH, USA). The amount of 10 ng of RNA was reverse transcribed using a TaqMan™ MicroRNA Reverse Transcription Kit (Applied Biosystems; Foster City, CA, USA) and specific primers for miR-21, miR-34a, miR-146a and miR-204 (Applied Biosystems; Foster City, CA, USA). Briefly, for reverse transcription reactions, 10 ng of total RNA was prepared with 3 μL of each of microRNA primers (100 mM), 0.15 μL of 100 mM dNTPs, 1.5 μL of 10X RT buffer, 0.2 μL of 20 units/μL RNase-inhibitor, 1 μL of MultiScribe Reverse Transcriptase (50 units/μL) and 4.15 μL of DEPC treated water in a reaction volume of 15 μL. The RT protocol was as follows: 16 °C for 30 min, 42 °C for 30 min and 85 °C for 5 min. Following reverse transcription, quantitative PCR was performed using TaqMan Fast Advanced Master Mix (Applied Biosystems; Foster City, CA, USA) and oligos from the same assay kits of Applied Biosystems as specified before. Values were normalized with the U6 snRNA as the reference gene (Applied Biosystems; Foster City, CA, USA). According to manufacturer’s guidelines, the relative quantity of each micro-RNA was determined by ∆∆C_T_ method [58].

#### 2.3.13. Immunohistochemistry

As previously described [53], a F4/80 immunohistochemistry was used to assess macrophage infiltration in adipose tissue and gastrocnemius muscle. Briefly, gastrocnemius and epididymal WAT samples were fixed in phosphate-buffered saline (PBS) with 4% paraformaldehyde (PFA) overnight. Sections of 5 μm were cut in a microtome and mounted on Superfrost slides. Sections were dried overnight at 37 °C, deparaffinized with xylol and rehydrated with distilled water. The endogenous peroxidase activity was abolished using 3% H_2_O_2_. Then, sections were warmed up 5 min with citrate buffer for antigen retrieval. After washing and blockade with PBS + 5% BSA for 1 h, sections were incubated overnight at 4 °C with the primary antibody (F4/80 (GTX101895, GeneTex, Irvine, California, USA); 1:100). Afterwards, sections were incubated 25 min with the corresponding biotin-conjugated secondary antibody (Thermo Fisher Scientific, Hampton, NH, USA; 1:1000) after a wash. Then, streptavidin conjugated with peroxidase (Thermo Fisher Scientific, Hampton, NH, USA) was added and diaminobenzidine (DAB) was used to revealed antibody signal. Hematoxylin was used to counterstain the nuclei for 1 min.

The number of macrophages was estimated by dividing the total number of positive cells by the total analyzed area. A 100× 1.25 NA objective of a light microscope (Leica, Wetzlar, Germany) was used and images were acquired using a microscope fitted with a 50 × 0.8 NA objective.

#### 2.3.14. Statistical Analysis

Values are expressed as means ± standard error of the mean (SEM). Results from stability experiments were analyzed by Student’s t-test and the results from the in vivo experiment were analyzed by one-way ANOVA followed by the Bonferroni post-hoc test using GraphPad Prism 5.0. (San Diego, CA, USA). A *p* value of <0.05 was considered significant.

## 3. Results

### 3.1. Study of the Oxidative Stability of the Oil Mixture in the Presence or Absence of the OLE

#### 3.1.1. Changes in Oxidation Parameters

The peroxide value increased in both AA:EVOO (*p* < 0.05) and AA:EVOO + OLE (*p* < 0.01) products from the third month compared to their initial values. However, there was a higher increase in the peroxide value in the absence of the OLE from 9 months on (*p* < 0.05). There was an increase in the anisidine value at the AA:EVOO from the third month of storage (*p* < 0.05), while the addition of the OLE prevented the significant increase in this parameter as it is always at lower levels than the AA:EVOO (*p* < 0.05) (Figure 1A,B).

#### 3.1.2. Changes in Fatty Acid Composition

There were no changes in the content of SFA and MUFA in both AA:EVOO and AA:EVOO + OLE products during the storage process. Percentage of PUFA in both products was decreased after 6 months of storage compared to the beginning (*p <* 0.05 for AA:EVOO, *p* < 0.001 for AA:EVOO +OLE), although its content remained higher in the presence of the OLE at this time point (*p* < 0.05) (Figure 1C–E).

#### 3.1.3. Changes in Polyphenolic Content

Changes in polyphenolic content as percentage compared to their initial concentration (Table 1) during storage are shown in Figure 1F–H.

Simple secoiridoids increased in the AA:EVOO mixture after 6 months of storage (*p* < 0.05). On the contrary, their concentration was significantly decreased in AA:EVOO + OLE after 3 months of storage (*p* < 0.05) (Figure 1F).

Levels of hydroxytyrosol and tyrosol derivatives decreased in the AA:EVOO mixture from the first month of storage on (*p* < 0.05), with this reduction being higher over time (*p* < 0.01). On the contrary, the levels of hydroxytyrosol and tyrosol derivatives in AA:EVOO + OLE did not change during the storage process (Figure 1G,H).

### 3.2. In Vivo Study

#### 3.2.1. Body Weight and Food Intake

At the end of the treatment period, young rats gained weight whereas old rats lost weight regardless of whether they had been treated with the nutraceutical or not. Caloric intake was unchanged between young and untreated old rats, but it was significantly reduced in old rats treated with the nutraceutical compared to both young (*p* < 0.001) and old untreated animals (*p* < 0.01) (Table 2).

#### 3.2.2. Organ Weights

The relative organ weights from young and old rats treated with vehicle or with the oil mixture and the OLE are shown in Table 2. Compared to young animals, old rats showed increased relative weights of spleen (*p* < 0.05), lumbar subcutaneous adipose tissue (*p* < 0.001) and epidydimal visceral adipose tissue (*p* < 0.001); and a decrease in liver (*p* < 0.001), soleus (*p* < 0.001) and gastrocnemius (*p* < 0.001) muscles. Supplementation with the oil mixture + OLE prevented the age-induced increase in the relative weights of epididymal adipose tissue (*p* < 0.05) and spleen (*p* < 0.05); and the age-induced decrease in the relative weights of the soleus (*p* < 0.05) and gastrocnemius (*p* < 0.05) muscles. Compared to untreated old animals, the treatment also increased the relative weight of the interscapular brown adipose tissue (*p* < 0.05).

#### 3.2.3. Blood Pressure

Old rats showed increased MAP compared to young animals (*p* < 0.05). Treatment with the nutraceutical product prevented the age-induced increased in the MAP (*p* < 0.05) (Table 3).

#### 3.2.4. Lipid Profile, Serum Levels of Metabolic Hormones and HOMA-IR Index

Aging was associated with a significant increase in the circulating levels of total lipids (*p* < 0.05), triglycerides (*p* < 0.05), total cholesterol (*p* < 0.05), LDL-cholesterol (*p* < 0.01), insulin (*p* < 0.01), leptin (*p* < 0.01) and adiponectin (*p* < 0.001). Likewise, the HOMA-IR was also higher in older rats than in young rats (*p* < 0.05). Supplementation with the oil mixture + OLE prevented the age-induced increase in HOMA-IR (*p* < 0.05), total lipids (*p* < 0.01), triglycerides (*p* < 0.05), total (*p* < 0.01) and LDL-cholesterol (*p* < 0.001) and further increased the serum levels of adiponectin (*p* < 0.01) (Table 3).

#### 3.2.5. Serum Inflammatory Parameters

The circulating levels of IL-6 and TNFα were significantly increased in old rats compared to young rats (*p* < 0.05 for IL-6 and *p* < 0.01 for TNFα). However, the serum concentrations of these two cytokines were significantly reduced in old rats supplemented with the nutraceutical (*p* < 0.05 for both) (Table 3).

#### 3.2.6. Serum Micro-RNA Levels

Old rats treated with vehicle showed an increase in the circulating levels of miR-21, miR-34a and miR-146a compared to young rats (*p* < 0.05 for all). Treatment with the oil mixture + OLE did not modify the serum concentrations of miR-34a but it prevented the age-induced increase in the levels of miR-21 (*p* < 0.05) and miR-146a (*p* < 0.01) (Table 3).

#### 3.2.7. Serum Fatty Acids Percentage

Significant increase in the percentage of SFA and a reduction in the percentage of MUFA (*p* < 0.01 for both) were found in the serum old animals. Treatment with the nutraceutical product did not modify the percentage of SFA or MUFA but it significantly increased the percentage of PUFA compared to both young (*p* < 0.01) and old (*p* < 0.05) rats administered with the vehicle (Table 4).

Old rats showed increased serum concentrations of palmitic and palmitoleic acids (*p* < 0.05 for both) and reduced circulating levels of ALA and oleic acid (*p* < 0.01 for both). Supplementation with the nutraceutical did not modify the serum levels of palmitoleic and oleic acids but it prevented the age-induced increase in the serum concentrations of palmitic acid (*p* < 0.05). In addition, the circulating levels of stearic acid were significantly lower in old rats treated with the nutraceutical (*p* < 0.05) and the levels of both EPA and DHA were significantly higher compared to those of old untreated rats (*p* < 0.01 for both) (Table 4).

#### 3.2.8. Hepatic Gene Expression of Metabolic, Inflammatory and Oxidative Stress Markers and PGC-1α Levels

Both the hepatic mRNA levels of the enzymes GCK and GSK3β and the protein ratio p-GSK3β/GSK3β were significantly decreased by aging (*p* < 0.05) (Figure 2A,B). Treatment with the oil mixture + OLE increased GCK expression (*p* < 0.01) and prevented the age-induced decrease in p-GSK3β/GSK3β.

In the liver, aging was associated with an upregulation of the mRNA levels of the pro-inflammatory markers iNOS (*p* < 0.01) and TNFα (*p* < 0.05), whereas those of COX-2 (*p* < 0.05), IL-10 (*p* < 0.05), GPX (*p* < 0.05), GSR (*p* < 0.01), SOD-1 (*p* < 0.001), NOX-4 (*p* < 0.001) and Alox5 (*p* < 0.01) were significantly reduced (Figure 2C,D). Treatment with the nutraceutical attenuated the age-induced changes in the mRNA levels of TNF-α (*p* < 0.05), GSR (*p* < 0.05), SOD-1 (*p* < 0.01) and NOX-4 (*p* < 0.05). Moreover, hepatic gene expression of IL-6 was significantly lower in old rats supplemented with the nutraceutical compared to that of both young and untreated old rats (*p* < 0.05 for both).

#### 3.2.9. Peripheral Response to Insulin

Incubation of gastrocnemius muscle and epididymal adipose tissue explants with insulin increased the protein level ratio of p-Akt/Akt in samples from young and old rats treated with the oil mixture + OLE (*p* < 0.05 for all), but not in old rats administered with the vehicle (Figure 3)(Appendix A).

#### 3.2.10. Macrophage Infiltration in the Gastrocnemius and Epididymal Adipose Tissue

Old animals showed increased macrophage infiltration in the gastrocnemius and in the epididymal white adipose tissue (*p* < 0.001 for both). The 21 day treatment with the nutraceutical compound prevented the aging induced increase in macrophage infiltration in both tissues (*p* < 0.001 for both) (Figure 3C,D).

#### 3.2.11. Gene Expression of Inflammatory and Oxidative Stress Markers in the Gastrocnemius Muscle

The mRNA levels of the proinflammatory markers COX-2 (*p* < 0.05), IL-6 (*p* < 0.01) and IL-1β (*p* < 0.05) were increased in gastrocnemius muscle of old rats compared to young ones and unchanged by the treatment. However, the muscle gene expression of IL-10 was decreased by aging (*p* < 0.05) and this decrease was prevented by the treatment with the oil mixture + OLE (*p* < 0.05) (Figure 4A). Moreover, aging was associated with an overexpression of the enzyme GPx (*p* < 0.001) and a decrease in SOD-1 and NOX-1 in gastrocnemius muscle expressions (*p* < 0.05 for both). Nutraceutical treatment partially prevented the age-induced increase in GPx mRNA levels (*p* < 0.05) and significantly increased the gene expression of GSR and NOX-4 compared to both young (*p* < 0.01 and *p* < 0.05 respectively) and old (*p* < 0.05 for both) rats treated with vehicle (Figure 4B).

#### 3.2.12. Gene Expression of Sarcopenia-Related Markers in Gastrocnemius Muscle

Old rats showed an increased gene expression of HDAC-4 (*p* < 0.001), MyoD (*p* < 0.01), myogenin (*p* < 0.001), atrogin (*p* < 0.05) and MuRF1 (*p* < 0.01) and decreased mRNA levels of MHC-I (*p* < 0.05) and MHC-IIa (*p* < 0.01) compared to young rats. The treatment with the oil mixture and the OLE partially prevented the age-induced increase in the mRNA levels of HDAC-4 (*p* < 0.05), MyoD (*p* < 0.05) and myogenin (*p* < 0.01) and prevented the changes in atrogin, MuRF1 and MHC-IIa associated with aging (*p* < 0.05 for all) (Figure 5A).

In addition, aging was associated with a downregulation in the gene expression of PGC-1α in the gastrocnemius (*p* < 0.05) that was prevented after treatment with the nutraceutical (*p* < 0.05) (Figure 5B).

#### 3.2.13. Gene Expression of Inflammatory, Oxidative Stress and Metabolic Markers in the Epididymal White Adipose Tissue

The changes in the mRNA levels of inflammatory, oxidative stress and metabolic markers in the epididymal white adipose tissue are shown in Figure 6.

Aging was associated with a decreased gene expression of COX-2 (*p* < 0.001), IL-10 (*p* < 0.05), GSR (*p* < 0.01), SOD-1 (*p* < 0.05), NOX-1 (*p* < 0.01), NOX-4 (*p* < 0.05) and Alox5 (*p* < 0.01) and with an overexpression of IL-1β (*p* < 0.001) in the WAT. Supplementation with the oil mixture + OLE prevented the age-induced changes in the mRNA levels of IL-1β (*p* < 0.01) and IL-10 (*p* < 0.05) and significantly reduced the TNF-α mRNA levels compared to untreated old rats (*p* < 0.05) (Figure 6A,B).

In old rats, the gene expression of the lipid metabolic enzymes LPL (*p* < 0.01), HSL (*p* < 0.05) and FASN (*p* < 0.05) were reduced, whereas the gene expression of PPARγ was significantly increased (*p* < 0.05) compared to young rats (Figure 6C). The treatment with the oil mixture + OLE decreased the gene expression of LPL compared to old rats treated with vehicle (*p* < 0.05) and prevented the age-induced changes in mRNA levels of PPARγ (*p* < 0.05).

#### 3.2.14. Aortic Vasoconstriction

Arterial vasoconstriction in response to KCl (A) and norepinephrine (B) are shown in Figure 7A,B. Treatment with the nutraceutical prevented the age-induced decrease in arterial vasoconstriction in response to potassium chloride 100 mM (*p* < 0.05) and the age-induced increased arterial vasoconstriction (*p* < 0.01) in response to the increasing concentrations of norepinephrine (*p* < 0.05).

#### 3.2.15. Endothelium-Dependent and Independent Aortic Relaxation

While the endothelium-independent relaxation in aorta segments in response to sodium nitroprusside (NTP) was not modified among experimental groups (data not shown), the endothelium-dependent relaxation, which is measured as a response to acetylcholine, was significantly reduced in old rats (*p* < 0.01). Nutraceutical treatment prevented this alteration (*p* < 0.01) (Figure 7C).

#### 3.2.16. Aortic Response to Insulin

In all experimental groups, insulin induced dose-dependent relaxation. However, this relaxation was significantly reduced in old rats (*p* < 0.05) and improved by treatment with the oil mixture + OLE (*p* < 0.05) (Figure 7D). Moreover, release of nitrites to the culture medium was increased with insulin incubation in aorta segments from young (*p* < 0.05) and old rats treated with the nutraceutical (*p* < 0.05), but not in old rats treated with the vehicle (Figure 7E).

#### 3.2.17. Aortic Gene Expression of Inflammatory and Oxidative Stress Markers

The mRNA levels of inflammatory and oxidative stress markers in aortic tissue are shown in Figure 8A,B, respectively.

An increase in the gene expression of COX-2 (*p* < 0.001) and NOX-4 (*p* < 0.001) and a reduction in the mRNA levels of TNFα (*p* < 0.05), IL-6 (*p* < 0.01), GPx (*p* < 0.01), GSR (*p* < 0.05) and NOX-1 (*p* < 0.05) was found in old rats. Treatment with the nutraceutical attenuated the age-induced changes in COX-2 (*p* < 0.01) and NOX-4 (*p* < 0.05) mRNA levels and significantly decreased the gene expression of Alox5 compared to young animals (*p* < 0.01).

#### 3.2.18. Cardiac Gene Expression of Inflammatory and Oxidative Stress Markers

The were no changes in the cardiac inflammation markers between young and old rats administered with the vehicle. However, supplementation with the oil mixture + OLE downregulated the gene expression of COX-2 (*p* < 0.05 and *p* < 0.01) and IL-1β (*p* < 0.05 for both) compared to both untreated young and old rats, the gene expression of iNOS compared to old rats (*p* < 0.05) and the gene expression of IL-10 compared to untreated young rats (*p* < 0.05) (Figure 9A).

Old rats showed lower mRNA levels of the antioxidant enzymes GSR (*p* < 0.01) and SOD-1 (*p* < 0.001) and higher mRNA levels of the pro-oxidant NOX-1 (*p* < 0.05) compared to young rats. Treatment with the nutraceutical prevented the age-induced decrease in the mRNA levels of NOX-1 (*p* < 0.05) and reduced the gene expression of Alox5 compared to young rats (*p* < 0.05). NOX-4 mRNA levels were reduced in old rats regardless of whether they had been treated (*p* < 0.001) or not (*p* < 0.01) with the oil mixture and the OLE (Figure 9B).

## 4. Discussion

The aims of this study were assessing whether the addition of an OLE to a mixture between AO and EVOO improves the metabolic and cardiovascular alterations associated with aging and increases the oxidative stability of the fatty acids in the mixture. Our results show that the addition of OLE delayed primary peroxidation for 3 months and reduced the formation of secondary oxidation products after 12 months of study. Likewise, other authors have also demonstrated the successful incorporation of OLE to different types of oils, such as olive and palm oils [19] or sunflower oil [18], improving their oxidative stability. Moreover, OLE addition has been proven to be a good strategy to extend the shelf life of other foods different from oils such as milk, where it exerts antibacterial effects and reduces the loss of both fat and lactose up to ten days [59]. This oxidative improvement is most likely related to the presence of phenolic compounds such as hydroxytyrosol, the most concentrated phenolic found in this OLE [60], which is reported to exert antimicrobial effects and to increase the oxidative stability of oils [61] and meat [62].

Regarding the biological effects of this new nutraceutical product, previous studies from our group have shown that the separate administration of both OLE and the mixture between EVOO and AO prevents skeletal muscle atrophy [63,64] and attenuates some of the cardiometabolic alterations associated with aging in rats [46,60]. Thus, in this study we wanted to analyze whether these ingredients may act synergistically so that their co-administration could provide further benefits in the elderly. Unlike the separate administration of OLE and AA:EVOO, our results show that the co-administration of both ingredients to old rats for 3 weeks significantly reduced visceral adiposity, an effect that seems to be mediated by a decrease in adipogenesis since the mRNA levels of PPARγ were downregulated in epidydimal adipose tissue of treated rats. This effect is most likely due to the presence of OLE, which is reported to reduce adipogenesis through a decrease in PPARγ levels both in vitro [65] and in vivo [66,67]. Moreover, OLE administration to aged rats for 3 weeks also downregulated the gene expression of PPARγ in visceral adipose tissue [63], but this effect was insufficient to significantly decrease fat mass. Thus, the mixture of the two ingredients (Oil mixture + OLE) promote a decrease in adiposity that is not present when they are administered separately, suggesting a synergic effect.

In addition, treatment with the oil mixture + OLE also attenuated the age-induced increase in the serum concentrations of total lipids, triglycerides and total and LDL-cholesterol. These effects seem to be mediated by AO:EVOO since the administration of the oil mixture alone exerts the same effects [46], whereas OLE administration only reduces the circulating levels of LDL-c [60]. Within the oil mixture, EVOO is possibly responsible for the decrease in LDL-c [68] and n-3 PUFAs may account for the reduction in triglyceride levels, as it has been previously reported both in experimental animals [69,70] and in humans [71,72].

Similar to the lipids, the serum levels of leptin and adiponectin were also increased in old rats, which is correlated with the higher amount of body fat. Treatment with the nutraceutical did not modify the circulating levels of leptin, but it further increased the adiponectin serum concentrations. This effect is most likely mediated by OLE because its administration to old rats alone also promotes a significant increase in adiponectin levels [60], although this increase was not translated into a decreased HOMA-IR. However, in the present study the increased adiponectin levels are also associated with a significant reduction in the HOMA-IR, suggesting improved insulin sensitivity in aged rats supplemented with the nutraceutical compared to untreated animals. Likewise, other authors have reported increased adiponectin levels and reduced insulin resistance in animals supplemented with OLE [67,73]. The increased HOMA-IR in old untreated rats was associated with insulin resistance in insulin-dependent tissues such as skeletal muscle and visceral adipose tissue where insulin failed to activate the PI3K/Akt pathway as it did in young rats. Moreover, in the liver of old untreated rats, insulin resistance was observed by a reduced expression or activation of GCK and GSK3. All these alterations were attenuated in old rats supplemented with the nutraceutical, with these positive effects on insulin sensitivity in metabolic tissues being most likely related to the antioxidant and anti-inflammatory effects of both the oil mixture [46] and OLE [60,63]. Indeed, both ingredients have demonstrated its ability to decrease the circulating levels of IL-6 in old rats [46,60] as well as the gene expression of several inflammatory and oxidative stress markers in metabolic tissues contributing to reduced insulin resistance [46,60,63,64]. Likewise, other authors have reported that both n-3 PUFAs [74] and olive derivatives reduce oxidative stress and inflammation through several mechanisms, which include the inhibition of NF-κB activity, MAPK, interleukin gene expression and HIF-1α [75].

In the present study the co-administration of the oil mixture + OLE significantly reduces the circulating levels of TNF-α, which may be related to the increased adiponectin concentrations and the decreased HOMA-IR, since it is reported that HOMA-IR is inversely related to adiponectin and positively related to TNFα plasma levels [76]. Moreover, the combined administration of the ingredients reduced the expression of TNF-α and IL-6 and increased the expression of GSR and SOD-1 in the liver. In visceral adipose tissue, anti-inflammatory effects were also found since the treatment reduced macrophage infiltration and the gene expression of some inflammatory markers such as TNF-α and IL-1β. However, the expression of antioxidant enzymes was not significantly elevated in old-treated animals, which suggests that the antioxidant effects of the ingredients are tissue-dependent. Likewise, OLE administration to aged rats significantly reduced oxidative stress in the liver but not in other organs such as the brain or the heart [77], which suggests that, possibly due to its detoxifying role, the liver is more sensitive to the antioxidant effects of certain ingredients of natural origin than other organs. Polyphenols derived from olive products may be responsible for these beneficial effects since they are reported to improve metabolic insulin sensitivity and alleviate some of the manifestations of metabolic syndrome, including arterial hypertension and hyperglycemia [78].

A major finding of this study is that the administration of this new nutraceutical product to old rats significantly attenuates sarcopenia. Sarcopenia is defined as the loss of muscle mass associated with aging and has a high incidence and prevalence among old population [79]. Although both ingredients have been reported to increase gastrocnemius weight and to prevent the expression of inflammatory and atrophy-related markers in old rats such as IL-6 and HDAC-4 in the case of the oil mixture [46] and IL-6, IL-1β, HDAC-4 or myostatin in the case of OLE [63], in this study our results show that the combined treatment with AO:EVOO and OLE provides further benefit preventing not only the decrease in the weight gastrocnemius muscle but also in the weight of the soleus muscle. Furthermore, the co-administration of both ingredients attenuated the reduction in MHC-I and MHC-IIa and downregulated the gene expression of the atrogene MuRF-1, which plays a major role in the process of age-induced muscle loss [80]. Paradoxically, aging also increased the expression of myogenic markers such as MyoD and myogenin, possibly as a compensatory mechanism [81,82] and these effects were also attenuated by supplementation with the nutraceutical pointing to improved muscle condition in old-treated rats. The protective effects of the nutraceutical decreasing sarcopenia in old rats is possibly due to the presence of OLE, EEVO and n-3 PUFAs since all of them are reported to prevent this condition [83,84,85,86] and may be mediated by increased PPAR-α as this receptor is highly involved in muscle metabolism [87] and its activation is reported to decrease muscle atrophy [88,89] and improve muscle function in aged individuals [87]. Indeed, inhibition of PPAR-α expression is reported to abolish the protective effects of human fibroblast growth factor 19 (FGF19) on obesity-induced sarcopenia [90]. The increased mRNA levels of PPAR-α are also most likely related to the reduced insulin resistance found in the skeletal muscle of old rats treated with the nutraceutical since this transcription factor is reported to play a key role in muscle insulin sensitivity. Indeed, the processes of insulin resistance and sarcopenia are intimately related. A positive correlation between insulin sensitivity and muscle mass has been found in aged rats [38] and the blockade of sarcopenic markers in old mice is reported not only to prevent sarcopenia but also to increase insulin sensitivity [91]. As stated above, age-induced insulin resistance is highly related to both inflammation [43] and oxidative stress [92]. Thus, although most of the inflammatory and oxidative stress markers were unchanged in skeletal muscle of old rats supplemented with the oil mixture + OLE, the increased activation of the PI3K/Akt pathway in response to insulin in gastrocnemius explants is most likely related to the decreased macrophage infiltration and to the increased gene expression of the antioxidant enzyme GSR and the anti-inflammatory cytokine IL-10.

In addition to the beneficial effects on age-induced muscle alterations and metabolism, the results of this study show that the treatment with this new nutraceutical also exerts beneficial effects on vascular function. Particularly, it improved endothelial function increasing relaxation to acetylcholine and insulin and prevented the decreased arterial contraction to KCl in old animals. Furthermore, the age-induce increased vasoconstriction of aorta segments in response to noradrenaline. These changes are also present when both ingredients are administered alone [46,60] and may underly the reduction in arterial blood pressure found in old animals supplemented with the nutraceutical product. Thus, this treatment may be useful for managing arterial hypertension, which is a common feature in aged patients. The antihypertensive effects of the oil mixture + OLE may be due to the presence of n-3 PUFAs since previous studies have reported that high doses n-3 PUFA significantly decreases blood pressure in old and hypertensive subjects [93] thanks to their endothelium dependent and independent vasodilating effects [94]. Likewise, both EVOO and OLE may also contribute to the antihypertensive effects of the nutraceutical since both of them are reported to be effective in lowering blood pressure in both experimental animals [95,96] and in humans [97,98,99,100].

Finally, an important finding of this study is that treatment with this new nutraceutical product attenuates the age-induced alterations in the circulating concentrations of certain micro-RNAs (miR) related with age-induced co-morbidities such as miR-21 and miR-146a. miR-21 has been identified as a new circulating marker of inflammaging [101] and of renal and cardiac alterations in elderly patients [102]. It is associated with the release of pro-senescence signals, affecting DNA methylation and cell replication; its levels are decreased in healthy centenarians [101] and increased in patients suffering from cardiovascular diseases. miR-146a circulating levels correlate with age [103], are related to age-induced bone loss [104] and have been suggested as a biomarker of sarcopenia [105]. Thus, the improvement in the circulating levels of these two micro-RNAs in old rats treated with the oil mixture + OLE may inform us about their better health status compared to untreated animals.

In summary, supplementation with a new nutraceutical based on mixture of AO, EVOO and OLE is effective against the impairment of the metabolic, cardiovascular and muscular conditions induced by aging thanks to its anti-inflammatory and antioxidant effects. It is important to point out that, even though both ingredients also exerted protective cardiometabolic effects when they were administered separately in a different set of animals, the co-administration of both provides further benefits decreasing visceral adiposity, TNF-α circulating levels, HOMA-IR and increasing soleus weight. In addition, the new nutraceutical increases the stability and reduces the oxidation of the n-3 PUFAs present in the AO.

## 5. Conclusions

In conclusion, this new nutraceutical product based on a mixture of EVOO, AO and OLE is stable from the oxidative point of view and may constitute a good strategy to treat and/or prevent the metabolic, cardiovascular and muscle alterations associated with aging.

## Figures and Tables

**Figure 1 antioxidants-10-01066-f001:**
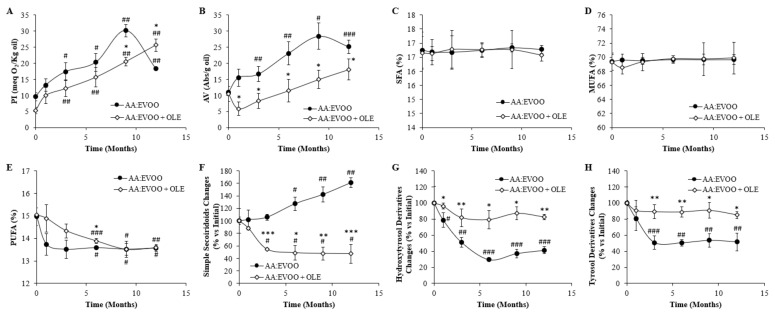
Peroxide index (PI) (**A**), anisidine value (AV) (**B**), saturated fatty acids (SFA; C16:0 + C18:0) (**C**), monounsaturated fatty acids (MUFA; C18:1n-9 + C18:1n-6) (**D**), polyunsaturated fatty acids (PUFA; EPA + DHA + DPA) (**E**), simple secoiridoids (**F**), hydroxytyrosol derivatives (**G**) and tyrosol derivatives (**H**) changes during storage in the oil mixture in presence/absence of the OLE. Values are represented as mean ± SEM. * *p* < 0.05 vs. AA:EVOO; ** *p* < 0.01 vs. AA:EVOO; *** *p* < 0.001 vs. AA:EVOO; # *p* < 0.05 vs. time 0; ## *p* < 0.01 vs. time 0; ### *p* < 0.001 vs. time 0. AA = algae oil; EVOO = extra virgin olive oil; OLE = olive leaves extract.

**Figure 2 antioxidants-10-01066-f002:**
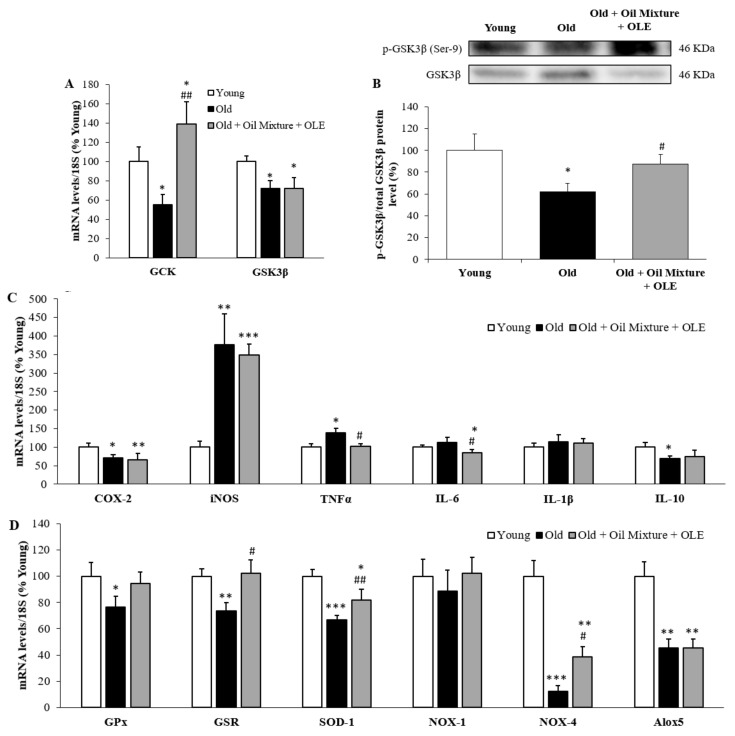
Hepatic mRNA concentrations of Glucokinase and Glycogen synthase kinase 3β (**A**), protein ratio between p-GSK3β and GSK3β (**B**); and mRNA concentrations of cyclooxigenase-2, inducible Nitric Oxide Synthase, Tumor Necrosis Factor α and Interleukin 6 and 1β (**C**) and of Glutathione Peroxidase and Reductase, Super Oxide Dismutase 1, NADPH oxidase 1 and 4 and Lipoxygenase (**D**) of young rats, old rats and old rats treated 21 days with the Oil Mixture and the OLE. Values are represented as mean ± SEM. * *p* < 0.05 vs. Young; ** *p* < 0.01 vs. Young; *** *p* < 0.001 vs. Young; # *p* < 0.05 vs. Old; ## *p* < 0.05 vs. Old.

**Figure 3 antioxidants-10-01066-f003:**
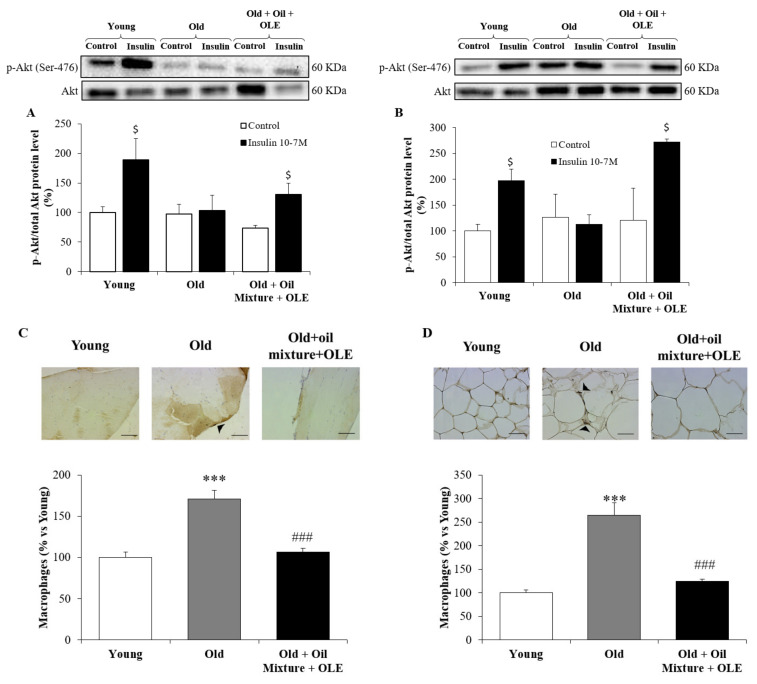
Ratio between protein levels of p-Akt and Akt in gastrocnemius (**A**) and epididymal visceral adipose tissue (**B**); and macrophages infiltration by immunohistochemistry in gastrocnemius (**C**) and epididymal visceral adipose tissue (**D**) of young rats, old rats and old rats treated 21 days with the Oil Mixture and the OLE. Values are represented as mean ± SEM. *** *p* < 0.001 vs. Young; ### *p* < 0.001 vs. Old; $ *p* < 0.05 vs. Control. Black line in the immunohistochemistry pictures represents 50 µm.

**Figure 4 antioxidants-10-01066-f004:**
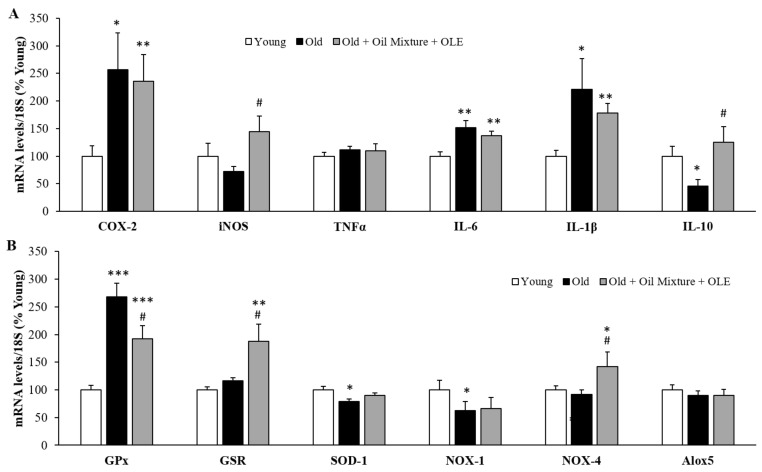
Gastrocnemius mRNA concentrations of cyclooxigenase-2, inducible Nitric Oxide Synthase, Tumor Necrosis Factor α and Interleukin 6, 1β and 10 (**A**); and of Glutathione Peroxidase and Reductase, Super Oxide Dismutase 1, NADPH oxidases 1 and 4 and Lipoxygenase (**B**) of young rats, old rats and old rats treated 21 days with the Oil Mixture and the OLE. Values are represented as mean ± SEM. * *p* < 0.05 vs. Young; ** *p* < 0.01 vs. Young; *** *p* < 0.001 vs. Young; # *p* < 0.05 vs. Old.

**Figure 5 antioxidants-10-01066-f005:**
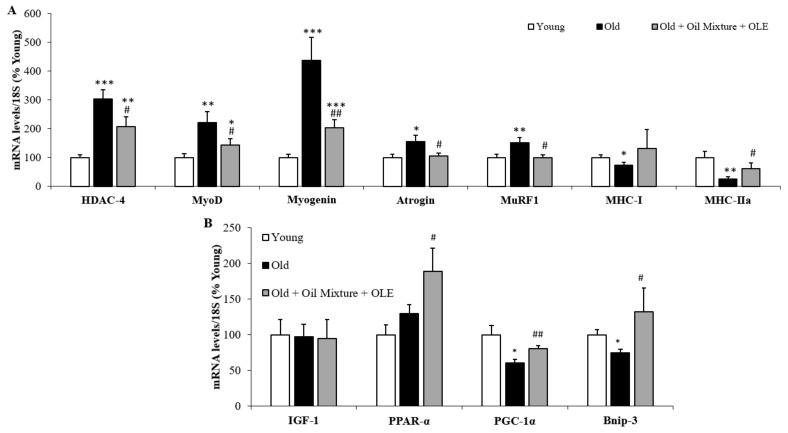
Gastrocnemius mRNA concentrations of Histone deacetylase 4, Myoblast determination protein 1, Myogenin, Atrogin, Muscle RING-finger protein-1, Myosin heavy chain I and myosin heavy chain IIa (**A**); and Insulin growth factor 1, peroxisome proliferator activated receptor α, PPARγ coactivator 1α and Bnip-3 (**B**) of young rats, old rats and old rats treated 21 days with the Oil Mixture and the OLE. Values are represented as mean ± SEM. * *p* < 0.05 vs. Young; ** *p* < 0.01 vs. Young; *** *p* < 0.001 vs. Young; # *p* < 0.05 vs. Old; ## *p* < 0.01 vs. Old.

**Figure 6 antioxidants-10-01066-f006:**
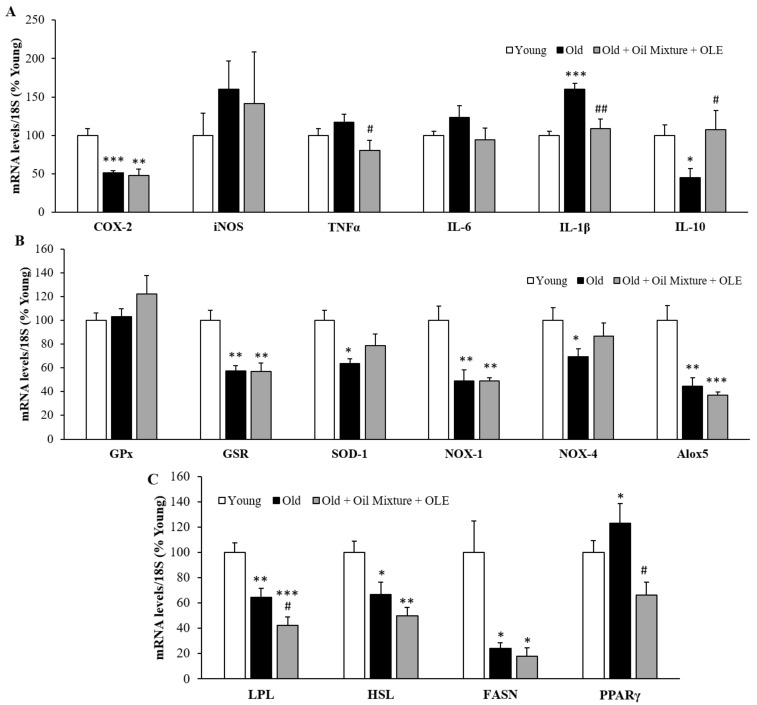
Epididymal adipose tissue mRNA concentrations of cyclooxigenase-2, inducible Nitric Oxide Synthase, Tumor Necrosis Factor α and Interleukin 6, 1β and 10 (**A**); and of Glutathione Peroxidase and Reductase, Super Oxide Dismutase 1, NADPH oxidases 1 and 4 and Lipoxygenase (**B**); and of Lipoprotein lipase, Hormone-sensitive lipase, Fatty acid synthase and Peroxisome proliferator activated receptor γ (**C**) of young rats, old rats and old rats treated 21 days with the Oil Mixture and the OLE. Values are represented as mean ± SEM. * *p* < 0.05 vs. Young; ** *p* < 0.01 vs. Young; *** *p* < 0.001 vs. Young; # *p* < 0.05 vs. Old; ## *p* < 0.05 vs. Old.

**Figure 7 antioxidants-10-01066-f007:**
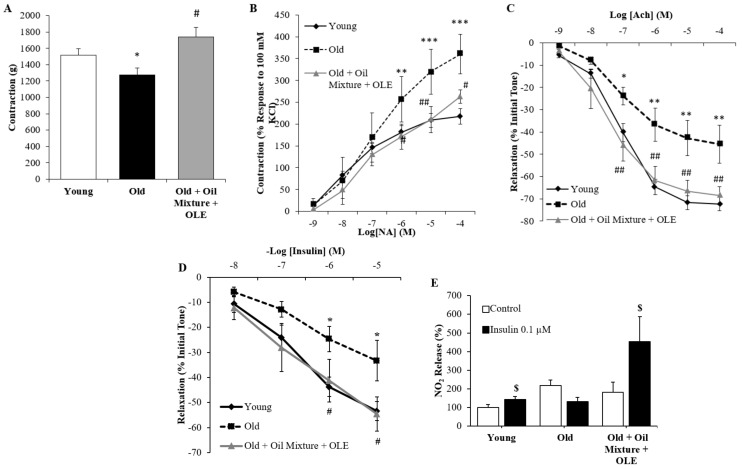
Contraction of abdominal aortic segments to potassium chloride 100 mM (**A**) and to norepinephrine (**B**); relaxation of thoracic aortic segments to acetylcholine (**C**) and to insulin (**D**); and nitrites released from aorta segments to culture medium in presence/absence of insulin (0.1 µM) (**E**) of young rats, old rats and old rats treated 21 days with the Oil Mixture and the OLE. Values are represented as mean ± SEM. * *p* < 0.05 vs. Young; ** *p* < 0.01 vs. Young; *** *p* < 0.001 vs. Young; # *p* < 0.05 vs. Old; ## *p* < 0.01 vs. Old; $ *p* < 0.05 vs. Control.

**Figure 8 antioxidants-10-01066-f008:**
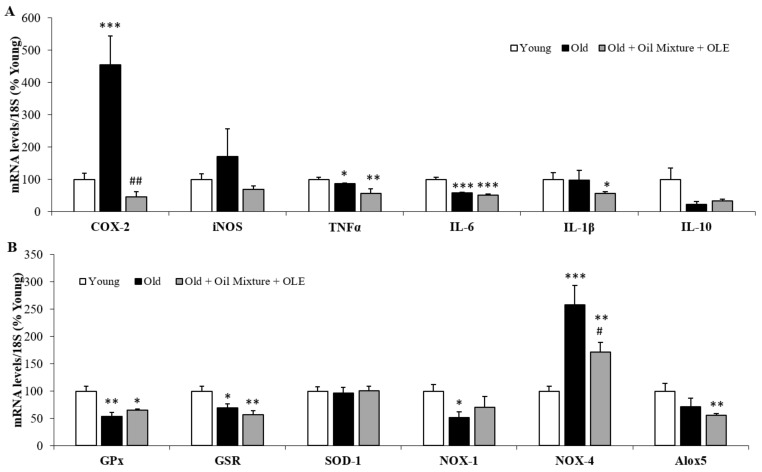
Aortic mRNA concentrations of cyclooxigenase-2, inducible Nitric Oxide Synthase, Tumor Necrosis Factor α, Interleukin 6, 1β and 10 (**A**); and Glutathione Peroxidase and Reductase, Super Oxide Dismutase 1, NADPH oxidases 1 and 4 and Lipoxygenase (**B**) of young rats, old rats and old rats treated 21 days with the Oil Mixture and the OLE. Values are represented as mean ± SEM. * *p* < 0.05 vs. Young; ** *p* < 0.01 vs. Young; *** *p* < 0.001 vs. Young; # *p* < 0.05 vs. Old; ## *p* < 0.01 vs. Old.

**Figure 9 antioxidants-10-01066-f009:**
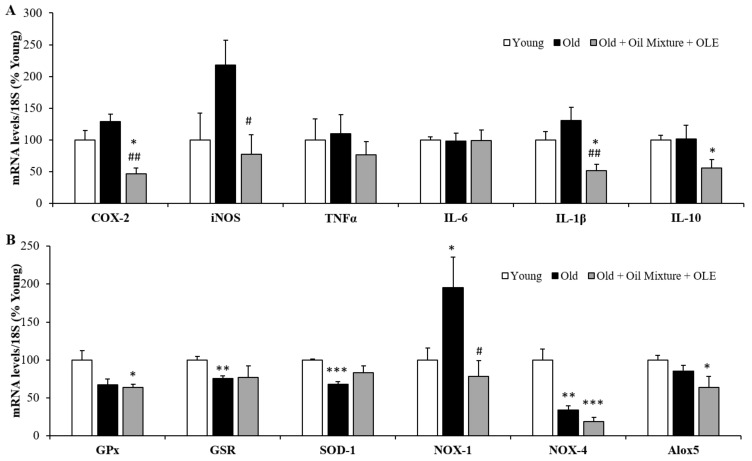
Cardiac mRNA concentrations of cyclooxigenase-2, inducible Nitric Oxide Synthase, Tumor Necrosis Factor α, Interleukin 6, 1β and 10 (**A**); and of Glutathione Peroxidase and Reductase, Super Oxide Dismutase 1, NADPH oxidases 1 and 4 and Lipoxygenase (**B**) of young rats, old rats and old rats treated 21 days with the Oil Mixture and the OLE. Values are represented as mean ± SEM. * *p* < 0.05 vs. Young; ** *p* < 0.01 vs. Young; *** *p* < 0.001 vs. Young; # *p* < 0.05 vs. Old; ## *p* < 0.01 vs. Old.

**Table 1 antioxidants-10-01066-t001:** Phenolic fraction (mg/g oil) of the oil mixture in presence/absence of the OLE.

	AA:EVOO	AA:EVOO + OLE
Simple secoiridoids	3.57 ± 0.2	400.1 ± 5.2 **
Hydroxytyrosol derivatives	11.7 ± 0.1	48.1 ± 1.9 *
Tyrosol derivatives	19.2 ± 0.2	58.5 ± 2.0 *

Data are represented as mean value ± SEM; *n* = 3 samples/oil. * *p* < 0.05 vs. AA:EVOO; ** *p* < 0.01 vs. AA:EVOO. AA = algae oil; EVOO = extra virgin olive oil; OLE = olive leaves extract.

**Table 2 antioxidants-10-01066-t002:** Body weight change, food intake and relative organ weights (mg/100 g body weight) of young rats, old rats and old rats treated with the oil mixture and the OLE.

	Young	Old	Old + Oil Mixture + OLE
Body weight change (g)	18.5 ± 1.8	−30.4 ± 5.7 **	−37.9 ± 4.5 *
Food intake (g/rat/day)	20.9 ± 0.7	17.1 ± 1.6	7.8 ± 0.9
Heart	314.6 ± 10.8	291.0 ± 14.4	274.0 ± 10.5
Epidydimal visceral adipose tissue	2107.1 ± 183.7	3201.8 ± 139.6 ***	2381.3 ± 317.9 *#
Lumbar subcutaneous adipose tissue	1063.1 ± 139.6	3974.2 ± 661.4 ***	4029.1 ± 161.8 ***
Interscapular brown adipose tissue	105.6 ± 9.8	111.1 ± 12.9	151.9 ± 11.2 ***#
Periaortic adipose tissue	39.3 ± 5.9	47.8 ± 7.5	49.9 ± 3.6 *
Kidneys	550.6 ± 5.9	512.1 ± 49.6	447.5 ± 21.2 ***
Suprarenal glands	13.5 ± 0.7	12.4 ± 1.2	10.4 ± 0.7 **
Liver	2932.8 ± 103.6	2204.2 ± 138.6 ***	2310.8 ± 119.5 ***
Spleen	151.6 ± 3.9	175.5 ± 14.6 *	142.3 ± 6.1 #
Soleus	39.2 ± 2.1	26.0 ± 1.6 ***	30.2 ± 1.2 ***#
Gastrocnemius	534.4 ± 12.1	329.4 ± 15.8 ***	367.2 ± 13.8 ***#

Data are represented as mean value ± SEM; *n* = 6–11 samples/group. * *p* < 0.05 vs. Young; ** *p* < 0.01 vs. Young; *** *p* < 0.001 vs. Young; # *p* < 0.05 vs. Old.

**Table 3 antioxidants-10-01066-t003:** MAP, lipid profile, hormone concentrations, inflammatory parameters and miRNA levels in the serum of young rats, old rats and old rats treated with the oil mixture and the OLE.

	Young	Old	Old + Oil Mixture + OLE
MAP (mmHg)	123.6 ± 3.0	140.1 ± 6.1 *	114.8 ± 1.8 #
Glycemia (mg/dL)	90.7 ± 3.8	72.4 ± 11.1	55.0 ± 7.9 ***
Total Lipids (mg/dL)	853.1 ± 63.9	1051 ± 37.4 *	793.3 ± 56.7 ##
Triglycerides (mg/dL)	97.6 ± 13.5	158.5 ± 36.4 *	63.5 ± 15.4 #
Total Cholesterol (mg/dL)	135.1 ± 15.3	199.3 ± 13.9 *	114.3 ± 13.6 ##
LDL-cholesterol (mg/dL)	28.8 ± 2.7	47.8 ± 2.6 **	17.82 ± 3.0 *###
HDL-cholesterol (mg/dL)	15.7 ± 0.6	13.4 ± 2.2	13.0 ± 1.5 *
Insulin (ng/mL)	17.7 ± 5.2	81.3 ± 32.2 **	19.9 ± 3.7 *#
HOMA-Index	1.75 ± 0.3	13.1 ± 5.4 *	2.59 ± 0.7 #
Leptin (ng/mL)	11.82 ± 1.4	30.2 ± 5.9 **	35.8 ± 7.6 **
Adiponectin (mg/dL)	67.1 ± 6.7	108.9 ± 4.3 ***	145.6 ± 7.5 ***##
Interleukin-6 (pg/mL)	135.6 ± 7.1	188.7 ± 24.1 *	130.1 ± 34.1 #
TNFα (pg/mL)	0.1 ± 0.1	1.8 ± 0.9 *	0.3 ± 0.2 #
miRNA-21/U6 (%)	100.0 ± 39.6	447.3 ± 151.3 *	155.6 ± 48.7 #
miRNA-34a/U6 (%)	100.0 ± 18.7	325.4 ± 96.3 *	304.8 ± 140.2
miRNA-146a/U6 (%)	100.0 ± 16.6	184.3 ± 43.8 *	51.4 ± 16.0 ##
miRNA-204/U6 (%)	100.0 ± 33.8	105.2 ± 38.3	101.7 ± 34.5

Data are represented as mean value ± SEM; *n* = 8–11 samples/group. * *p* < 0.05 vs. Young; ** *p* < 0.01 vs. Young; *** *p* < 0.001 vs. Young; # *p* < 0.05 vs. Old; ## *p* < 0.01 vs. Old; ### *p* < 0.001 vs. Old. MAP = Medial Arterial Pressure.

**Table 4 antioxidants-10-01066-t004:** Fatty acid proportion (%) of young rats, old rats and old rats treated with oil mixture and the OLE.

	Young	Old	Old + Oil Mixture + OLE
SFA	25.0 ± 0.8	28.4 ± 0.4 **	27.1 ± 1.2
MUFA	49.6 ± 0.6	42.0 ± 2.2 **	41.6 ± 1.3 ***
PUFA	15.2 ± 2.5	16.1 ± 1.2	27.4 ± 2.5 **##
Palmitic Acid (C16:0)	8.28 ± 0.3	9.92 ± 0.7 *	8.11 ± 0.1 #
Palmitoleic Acid (C16:1)	3.52 ± 0.5	5.47 ± 0.5 *	7.65 ± 1.5 *
Stearic Acid (C18:0)	13.2 ± 0.6	13.0 ± 0.5	11.4 ± 0.3 *#
Oleic Acid (C18:1)	47.7 ± 0.7	40.8 ± 1.8 **	40.6 ± 1.6 **
Linoleic Acid (C18:2)	1.84 ± 0.2	1.15 ± 0.4	1.07 ± 0.6
ALA (C18:3)	3.42 ± 0.2	2.68 ± 0.1 **	2.67 ± 0.5
EPA (C20:5n-3)	4.68 ± 1.2	6.47 ± 0.5	13.7 ± 1.4 *##
DHA (C22:6n-3)	7.12 ± 1.5	6.95 ± 0.7	11.3 ± 1.3 **##

Data are represented as mean value ± SEM; *n* = 6–11 samples/group. * *p* < 0.05 vs. Young; ** *p* < 0.01 vs. Young; *** *p* < 0.001 vs. Young; # *p* < 0.05 vs. Old; ## *p* < 0.01 vs. Old. ALA = α-linolenic acid; DHA = docosahexaenoic acid; EPA = eicosapentaenoic acid; MUFA = monounsaturated fatty acids; PUFA = polyunsaturated fatty acids; SFA = saturated fatty acids.

## Data Availability

Data is contained within the article.

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
