# Peer review of "Addition of Olive Leaf Extract to a Mixture of Algae and Extra Virgin Olive Oils Decreases Fatty Acid Oxidation and Synergically Attenuates Age-Induced Hypertension, Sarcopenia and Insulin Resistance in Rats"

_antioxidants, 2021, doi:10.3390/antiox10071066_

Round 1

Reviewer 1 Report

The manuscript entitled „Addition of olive leaf extract to a mixture of algae and extra vir-2 gin olive oils decreases fatty acid oxidation and synergically attenuates age-induced hypertension, sarcopenia and insulin resistance in rats“ written by González-Hedström et al. is a well written paper documenting the effect of a mixture of olive leaf extract (OLE), algae oil (AO) and extra virgin olive oil (EVOO) on age-related cardiometabolic changes in rats. The study is very complex and reports the impact of tested mixture on a wide range of parameters including biometric, cardiovascular, and metabolic, as well as potential molecular mechanisms behind beneficial effect of this mixture in multiple organs including muscle, liver, heart, and adipose tissue, especially those related to inflammation and oxidative stress. Moreover, the paper documents that addition of OLE to a mixture of AO&EVOO stabilizes its antioxidant properties in time of storage thus improving its ability to exert its beneficial effects in clinical application where storage conditions are of particular importance. The data presented in the paper are novel, scientifically sound, and have a good potential to be translated to clinical praxis. Finally, the topic of the study fits to the scope of the target journal. Thus, the paper is suitable to be published in Antioxidants; however, I have several minor comments and suggestions to improve the paper before publication:

  • Despite overall organization of the paper is very good and clear, I would suggest not interrupt the Discussion with Figures (8,9) but I recommend place all figures before the beginning of Discussion (move Figs 8 and 9).
  • Results are sound and complex; however, it is not clear to me why the WB measurements of protein levels are so inconsistent in comparison to mRNA measurements by PCR. In fact, at protein level authors show only p-Akt/Akt ratio in gastrocnemius and visceral tissue, and p-GSK3β/GSK3β ratio in the liver. There are no data of protein levels in heart or vessels; moreover, those in above mentioned tissues are in imbalance to very complex and consistent data of mRNA expressions by PCR. Could authors explain this imbalance? Were levels of other proteins found unchanged by WB? Or were there any methodological problems that did not allow authors provide more complex and consistent data from WB, similarly as in case of PCR data?
  • In Introduction authors list known positive effects of EVOO and Olive oil constituents, primarily due to their antioxidant properties, on cardiometabolic parameters (lines 101-117). Here, the recent comprehensive review paper on the role of antioxidants including Mediterranean diet compounds, also EVOO and OLE, in targeting oxidative stress in CMDs should be cited (PMID: 33905865).
  • In Methods/Treatment authors introduce group names for old rats (Old) but do not introduce group name for young rats. This is strange, and also in the text can be found that “Old” appears with capital letter as the group name but in the same sentence “young” appears just with small letter, e.g. page 9 line 395. In contrast, in Tables, Figures and Figure legends “Young” also appears in capital letter. This looks strange and authors should unify the group naming and use properly in the MS.
  • There are several abbreviation throughout the whole MS that are not properly introduced when firstly appear in the text, e.g. MAPK, GLUT, PUFA, ALA, EPA, DHA, MUFA, Alox, WAT, FASN, LPL, HSL, etc. Please, go through the text carefully and introduce all abbreviations when first appear in the text. In addition, I suggest create the List of Abbreviations for the paper to help readers in the orientation in the text.
  • In Introduction, please unify the abbreviation PI3-kinase/Akt versus PI3K/Akt, and explain the abbreviation PI3K as well. Also, please be consistent in all abbreviation form, e.g. NOX1 vs. NOX-1, or NOX4 vs. NOX-4. (one form appears in Methods, other in Figures and Results).
  • Authors declare that they measured primary and secondary oxidation compounds in the study, I would prefer using the term “oxidation products” rather than “oxidation compounds”
  • In Introduction, line 119, the citation of Swanson et al. 2012 is not numbered. Please correct this.
  • There are several typing errors in the paper, e.g. “ciclooxigenase” appears many times in the text as well as in the Figure legends – should be corrected to “cyclooxygenase”. In Abstract, double “with with” appears. Also, several spaces are missing between the words, e.g. line 281 “10min”, or line 657 “LDL-c[„.

Reviewer 2 Report

Minor grammar and spelling changes are shown in the attached file

Reviewer 3 Report

The authors investigated a new nutraceutical, AO+EVOO+OLE mixture, on lipid stability and the effect of supplementation on cardiac and metabolic tissues in aged rats for 21 days (3 weeks).

Strengths:

This study investigates the stability of a high PUFA nutraceutical with the addition of olive leaf extract over a period of a year measuring multiple oxidative, lipid, and polyphenolic compounds.  The stable mixture of AO+EVO+OLE mixture was then tested in vivo on old rats of various physiological factors (inflammation, insulin resistance, sarcopenia, contractility) in blood and multiple tissues: heart, aorta, skeletal muscle, liver and adipose tissue.  Lipids, miRNA, and mRNA of inflammatory markers and enzymes were measured in tissues.

Weaknesses:

Minor weaknesses are noted below.

General

  1. Unnecessary capitalization of words (example line 187).
  2. Sentences with numbers need to be spelled out. Example line 192.

Abstract

  1. Line 7: decreasing makers of inflammation and oxidative stress. Since you didn’t directly measure inflammation and oxidative stress in tissues, it needs to be clearly stated that you measured markers (mRNA of cytokines and oxidative-stress related enzymes) of inflammation and oxidative stress throughout the paper.

Introduction

  1. The introduction is very long; I suggest condensing and discussing the last three paragraphs first. Need to focus on the nutraceutical and then what you plan to measure.
  2. Some awkward sentences (line 40-41 for example) and incorrect citation style line 119.

Materials and Methods

  1. Need to briefly describe 2.2.4 and calculation of PV and AV.
  2. Need to elaborate on 2.3.8.

Results

  1. Minor typos need to be corrected: example line 355 and cyclooxygenase in figure legends.
  2. Body weights and food intake need to be included in table 2.
  3. Need to rephrase aging-induced. Do no use aging; three weeks of intervention is not long enough for aging, but comparison between aged or older animals compared to young can be compared.  I suggest using age-induced or age related,  aged, or older mice.
  4. Use percent instead of % symbol, line 438.
  5. Explain why miRNA values of young are all 100.0. How is this possible?
  6. Line 488; insulin as well as vehicle did not cause an increase in p-AKT in gastrocnemius and epididymal adipose tissue.
  7. Explain why all control animals do not have the same p-AKT induction between young, old, and old+oil mixture (figure 3A).
  8. Figure 3C and D-magnification the same and adipocytes larger?
  9. Insulin values in figure 7: change to micromolar instead of molar 10-7 M.

Discussion

  1. Correct typos (example line 656).
  2. Clarify the fact that you cannot make definitive assumptions about the separate compounds and the synergistic effects b/c it wasn’t determined in this study and can be speculative in comparing other studies with separated compounds.
